# The Impact of Epigenetic Modifications on Adaptive Resistance Evolution in Glioblastoma

**DOI:** 10.3390/ijms22158324

**Published:** 2021-08-03

**Authors:** Qiong Wu, Anders E. Berglund, Arnold B. Etame

**Affiliations:** 1Department of Neuro-Oncology, H. Lee Moffitt Cancer Center and Research Institute, 12902 Magnolia Drive, Tampa, FL 33612, USA; Qiong.Wu@moffitt.org; 2Department of Biostatistics and Bioinformatics, H. Lee Moffitt Cancer Center and Research Institute, 12902 Magnolia Drive, Tampa, FL 33612, USA; anders.berglund@moffitt.org

**Keywords:** epigenetic modifications, DNA methylation, histone methylation, histone acetylation, chromatin remodeling, glioblastoma, resistance

## Abstract

Glioblastoma (GBM) is a highly lethal cancer that is universally refractory to the standard multimodal therapies of surgical resection, radiation, and chemotherapy treatment. Temozolomide (TMZ) is currently the best chemotherapy agent for GBM, but the durability of response is epigenetically dependent and often short-lived secondary to tumor resistance. Therapies that can provide synergy to chemoradiation are desperately needed in GBM. There is accumulating evidence that adaptive resistance evolution in GBM is facilitated through treatment-induced epigenetic modifications. Epigenetic alterations of DNA methylation, histone modifications, and chromatin remodeling have all been implicated as mechanisms that enhance accessibility for transcriptional activation of genes that play critical roles in GBM resistance and lethality. Hence, understanding and targeting epigenetic modifications associated with GBM resistance is of utmost priority. In this review, we summarize the latest updates on the impact of epigenetic modifications on adaptive resistance evolution in GBM to therapy.

## 1. Introduction

Glioblastoma (GBM) is a highly lethal tumor that is refractory to the current therapeutic options. The highly invasive and infiltrative nature of GBM precludes complete eradication through surgical resection. Although post-surgical radiation therapy (RT) confers modest increase in overall survival, the addition of temozolomide (TMZ) significantly increases overall survival [1]. However, the multimodal combination GBM therapies of surgical resection, TMZ, and RT provide a median survival of only 15 months in patients with newly diagnosed GBM [1]. The prognosis is markedly dismal in patients with recurrent GBM, whereby a highly resistant phenotype limits the effectiveness of additional chemotherapy [2,3]. The current GBM therapeutic paradigm depends on adjuvant TMZ for durable tumor control that is often short-lived. Hence, acquisition of resistance to TMZ is a critical mechanism of treatment failures in GBM. Identifying and targeting critical mechanisms of resistance in GBM is necessary to impactfully improve survival in GBM.

Therapeutic resistance is a complex process driven by multiple mechanisms that result in survival adaptations in cancer cells [4,5,6]. Resistance can occur through both genetic and non-genetic mechanisms. For instance, genetic mutations have been shown to play a critical role in mediating therapeutic resistance to a range of standard and targeted chemotherapies in cancers including GBM [7,8,9,10,11]. However, it is also possible for cancer cells to acquire resistance despite the absence of genetic mutations or alterations in drug targets [12,13,14,15,16,17,18,19]. Epigenetic mechanisms, for instance, could contribute to changes in the genome that are independent of DNA alterations in cancer cells. GBM has distinct genetic and epigenetic signatures that dictate tumorigenesis [20,21,22,23,24]. Hence, a consideration of the epigenetic complexities of GBM resistance is essential.

Epigenetic mechanisms have been implicated in GBM tumorigenesis and resistance [25,26,27] (Figure 1). Established epigenetics modifications include DNA methylation, histone methylation/acetylation, chromatin post-translational modification, and non-coding RNAs modification [28,29] (Figure 1). Genes encoding histone methyltransferases, histone demethylases, and histone deacetylases represent critical regulators of epigenetic modifications. Dysregulation of epigenetic regulators can facilitate transcription of genes that promote tumorigenesis [30,31,32,33], and resistance [34]. Hence, modulation of the epigenome of cancer cells provides a novel therapeutic approach in overcoming treatment resistance.

Therefore, it is essential to understand and target epigenetic events that mediate GBM resistance and recurrence. The epigenetic landscape of GBM has been extensively studied [35,36,37], and several epigenetic modifications have been identified as potential therapeutic targets in GBM [38,39]. In this review, we will mainly focus on the impact of epigenetic modifications on GBM resistance. We will highlight discoveries on epigenetic modulation of GBM drug resistance. Understanding how epigenetic modifications impact drug resistance in GBM will permit rational targeting of epigenetic modifications in GBM.

## 2. Epigenetic Alteration Involved in Drug Resistance of GBM

Cancer development is a complex process that entails sequential changes in the genome and epigenome. These changes contribute to both tumor heterogeneity, plasticity, and result in alterations in gene expression through modifications of nucleotides and proteins without changes in DNA sequences. Further, cancer cells employ epigenetic mechanisms to regulate gene expression and function in response to stressors [40]. Epigenetic modifications of DNA and histones, together with changes in nucleosome composition and chromatin arrangement therefore serve as an extra layer of gene expression regulation. Epigenetic modifications, such as methylation and acetylation, are catalyzed by specific enzymes and proteins that modify the epigenome rather than genome. The regulatory machinery in epigenetics involves enzymes and proteins that are broadly classified into one of the following categories: the “writers”, the “readers”, and the “erasers” [41]. The “writers” are enzymes such as DNA methyltransferase, histone lysine methyltransferases, and histone acetyltransferases that are responsible for the addition of modifications such as functional methyl/acetyl groups. The “readers” are proteins or enzymes such as methyl CpG binding proteins and histone methylation/acetylation readers that recognize the presence of epigenetic modifications. The “erasers” are enzymes such as histone demethylases and histone deacetylases that erase modifications on DNA/lysine residues and histone proteins. It is now established that epigenetic regulation of modifications impacts upon both nucleosome repositioning and chromatin accessibility resulting in gene expression [41,42]. In the genome, euchromatin regions are loosely packed and easily accessible for transcriptional activation whereas heterochromatin regions are tightly packed and less accessible [42].

Cancer drug resistance accounts for approximately 90% of cancer-related mortality [43,44,45]. There is overwhelming evidence that epigenetic modifications represent a potential mechanism for rapid acquisition of drug resistance in cancer [34,46,47,48,49]. Similar to other cancers, GBM cells can acquire resistance to drug therapy through multiple mechanisms. In general, both intrinsic genetic resistance and treatment-induced resistance are common mechanisms through which GBM cells can attain survival adaptation (Figure 2). GBM cells with drug target genetic mutations that confer resistance to therapy will persist. In addition, a small population of tumor cells without prior genomic mutations can transcriptionally evolve into drug tolerant status and persist despite treatment (Figure 2). This plasticity towards adaptive resistance is largely influenced by epigenetic modifications [44,46]. In GBM, phenotypic plasticity to radiation therapy has been observed from the residual glioma stem cell populations that exist beyond the confines of resected tumor [50]. Since such residual glioma stem cells exist beyond the extent of surgically resected disease, they serve a nidus for recurrence and lethality in GBM.

## 3. Targeting Epigenetic Alteration

DNA methylation, histone modifications, chromatin remodeling, and long non-coding RNAs (lncRNAs) are the main mechanisms of epigenetic regulation of transcriptional activation in cancers such as GBM. Since epigenetic modifications drive tumorigenesis and therapeutic resistance, there is a potential clinical benefit in targeting epigenetic regulators in GBM patients (Figure 3). Several epigenetic agents including histone methyltransferase inhibitors, DNA methyltransferase inhibitors, histone deacetylase inhibitors, and other agents are currently being tested in GBM patients in clinical trials (Table 1).

### 3.1. DNA Methylation

DNA methylation is one of the best characterized epigenetic modifications whereby DNA methyltransferases (DMNTs) preferentially methylate the C-5, N-4, N-6, and N-7 sites of DNA [74,75]. GBM methylation alterations can manifest as genome-wide hypomethylation, gene-specific hypomethylation, or hypermethylation [76,77,78,79,80]. DMNTs catalyze the transfer of methyl group S-adenosylmethionine to the fifth carbon of cytosine residue and form 5-metylcytosine as part of the DNA methylation process. In GBM, intrinsic DNA methylation has a significant impact on response to TMZ, the standard of care chemotherapy. TMZ is an alkylating agent responsible for methylation of guanine in position N7 and O6, and Adenine in N3. The role of DNMTs with respect to TMZ sensitivity in GBM has not been fully explored. Zhou et al. investigated whether DNMTs expression was associated with TMZ sensitivity in GBM cells and elucidated the underlying mechanism [81]. DNA methyltransferase 1 (*DNMT1*) expression was found to be downregulated in TMZ resistant GBM cells, negatively correlated with miR-20a expression, and positively correlated with TMZ sensitivity [81]. However, these findings were reported using a single GBM cell line that might not fully recapitulate the cellular hierarchical heterogeneity GBM stem cells.

O-6-methylguanine-DNA methyltransferase (*MGMT*) is a critical DNA damage repair gene, responsible for repairing DNA lesions of O6 adducts created by alkylating chemotherapy agent such as TMZ. *MGMT* expression is highly regulated through promoter methylation whereby hypermethylation of the *MGMT* promoter results in epigenetic silencing of *MGMT* expression leading to enhanced clinical response to TMZ [82,83]. Similarly, GBM patients with unmethylated *MGMT* promoter status demonstrated a worse response to TMZ [84]. Hence *MGMT* promoter methylation serves as a prognostic surrogate for TMZ response in GBM [84]. However, strategies to target *MGMT* have failed to improve TMZ response and survival in GBM patients in clinical trials [85,86]. Accordingly, there are likely other epigenetic mechanisms besides *MGMT* promoter methylation that impact upon adaptive resistance to TMZ. For instance, epigenetic regulation of XIAP associated factor 1 (*XAF1*), a previously reported tumor suppressor was recently reported to mediate plasticity towards adaptive resistance in GBM to TMZ [87].

Moreover, chemotherapy agents such as TMZ can induce DNA methylation alterations throughout the entire genome through mechanisms that are not completely understood [88,89,90]. It is also not clear why drug-induced hypermethylation selectively leads to a resistant phenotype as opposed to a sensitive phenotype in cancer cells. Methylation alterations involving promoters that modulate drug efflux transporters, pro-apoptotic genes, and DNA damage repair genes are potential contributory mechanisms towards drug resistance [82,91,92,93]. Other potential mechanisms of resistance include methylation alterations at enhancer sites on the genome. Collective evidence from recurrent GBM tumor tissues and TMZ resistant patient-derived xenografts implicates activation of an enhancer region between marker of proliferation Ki-67 and *MGMT* promoters as a driver of TMZ resistance [94]. TMZ sensitivity was restored following deletion of this enhancer region even in GBM cells with high MGMT expressions [94].

Given the established link between DNA hypermethylation and drug resistance, it has been suggested that DNA demethylation provides a novel therapeutic avenue to enhance TMZ sensitivity. Hence, there have efforts to evaluate demethylating agents as anti-cancer drugs. It should be noted that demethylating agents can have widespread unwanted systemic effects given the lack of selectivity. Such agents have been used to reverse DNA hypermethylation related resistance in GBM [95,96,97]. There is further preclinical and clinical evidence that demethylating agents are more effective when used in combination with other cancer therapies [69].

### 3.2. Histone Modification

The role of histone modifications on gene regulation in GBM is well recognized. Histone modifications alter gene expression without changing the DNA sequences and are catalyzed by specific enzymes and related proteins at four core histone proteins: H2A, H2B, H3, and H4. The most common modifications include acetylation, methylation, phosphorylation, and ubiquitylation. Histone acetylation results in gene activation while histone methylation can result to either gene activation or gene repression depending on the histone protein core and amino acid residue [98]. Abnormalities in histone modification can facilitate transcription of genes that drive GBM propagation, contributing to therapeutic resistance [99,100,101]. For instance, enhancer of zeste homolog 2 (EZH2) is a histone methyltransferase in the polycomb repressive complex 2 (PRC2) that has been implicated in cancer propagation [102]. *EZH2* expression is high in GBM and negatively impacts GBM survival [103]. EZH2 mechanistically upregulates c-MYC expression [104] and increases signal transducer and activator of transcription 3 (STAT3) phosphorylation [105], leading to GBM tumorigenesis and making EZH2 a compelling epigenetic target.

Methylation of arginine residues is one of the most common post-translational modifications of histone proteins. Methylation of arginine residues disrupts protein–protein interactions and associated downstream cellular processes. Protein arginine methyltransferases (PRMTs) are critical enzymes responsible for adding methyl groups onto arginine residues within target proteins. Emerging evidence within the last decade suggests a strong nexus between aberrancies in PRMTs function and GBM tumorigenesis [106,107]. Hence, PRMT inhibitors are currently under development for preclinical and clinical trials in GBM patients. Protein arginine methyltransferase 5 (PRMT5) [108,109,110], and protein arginine methyltransferase 1 (PRMT1) [111] are members of the PRMT family proteins which are over-expressed in GBM and have a negatively impact upon patient survival. Genetic depletion of either *PRMT5* [110,112,113] or *PRMT1* [111] significantly inhibits tumor growth in intracranial orthotopic mouse xenograft models. Furthermore, small molecule inhibitors of PRMT5 can drive GBM stem-like cells into senescence [113]. Recently, it was identified that protein arginine methyltransferase 6 (PRMT6) is required for methylation of regulator of chromatin condensation 1 (*RCC1*) and further induce proliferation, stem-like properties and tumorigenicity of GBM stem cells (GSCs) [114]. Depletion of PRMT6 with a small molecule inhibitor could suppress *RCC1* arginine methylation and improve the cytotoxic activity of radiotherapy against GSCs in vitro and in vivo.

The lysine demethylase (KDM) genes are histone modification enzymes that also have a very important role in GBM resistance. KDM genes have been implicated in the dysregulation of senescence, apoptosis, and tumor progression [115,116]. Particularly, lysine demethylase 2B (KDM2B) [117] and lysine demethylase 1A (KDM1A) [118,119] propagate survival adaptation towards a stemness phenotype in GBM [117]. Depletion of either KDM2B [117] or pharmacological inhibition of KDM1A [118,119] could induce apoptosis in GBM. Lysine demethylase 5A (KDM5A), an H3K4 demethylase, was found to be overexpressed in GBM cells that were adaptively resistant to TMZ [120,121]. Importantly, inactivation of KDM5A can efficiently restore TMZ sensitivity in adaptively resistant GBM cells [120,121]. Inhibition of KDM5A with small molecule inhibitors such as the pan-KDM inhibitor JIB 04 and the selective KDM5A inhibitor CP1445 efficiently inhibited tumor growth in vitro and in vivo of TMZ-resistant GBM [120,121]. KDM6B is a histone H3K27 demethylase that was investigated as a potential target for GBM treatment recently [121,122]. Lysine demethylase 6B (KDM6B) was overexpressed in GBM tissues and treatment with KDM5B specific inhibitor GSK-J4 significantly improved survival in GBM models of diffuse intrinsic pontine gliomas. Furthermore, the combination of KDM5A inhibitor JIB 04 and KDM6B inhibitor GSK-J4 resulted in significant synergy and potency against TMZ resistant GBM cells [121].

The inter-dynamics between histone acetylation and deacetylation in maintaining a balanced state of acetylation is critical to the regulation of gene transcription. Acetylation and deacetylation are mediated by histone acetyltransferases (HAT) and histone deacetylases (HDAC), respectively. Acetylation occurs through addition of acetyl groups to H3 and H4 and weakens the interaction between the histone core and DNA leading to facilitated transcription. HDACs are overexpressed and mutated in various solid and hematologic malignancies and play key roles in tumorigenesis [123,124]. HDACs have been implicated in chemoradiation resistance through inhibition of DNA double-strand break repair [125,126]. Furthermore, HDAC-mediated resistance to TMZ is easily reversed with inhibitors of HDACs in GBM [127,128,129,130,131]. HDAC inhibitors target several processes that are integral to tumorigenesis including the induction of cell cycle arrest, differentiation, senescence, cell death, and inhibition of angiogenesis [132,133]. Several HDAC inhibitors have already been approved by the FDA including Vorinostat, Belinostat, Romidepsin, Belinostat, Valproic acid, and Panobinostat. Given the role of HDACs in GBM, HDAC inhibitors have been extensively studied in GBM clinical trials (Table 1). The results so far suggest that HDAC inhibitors such as Vorinostat have no impact on overall survival even when combined with chemotherapy [134]. Moreover, besides issues with efficacy, HDAC inhibitors have undesirable side effect profiles which remain a huge impediment [135,136,137] (Table 1). A possible avenue for consideration is the combination of HDACs and KDM inhibitors based on reports that pharmacological inhibition of KDM1A can sensitize GBM cells to HDAC inhibitors in-vivo [138]. Further studies are necessary to determine the optimal combinatorial strategies for HDACs that permit efficacy as well as safety in GBM. Targeting HDAC-mediated resistance in GBM remains an active area of investigation.

### 3.3. Chromatin Remodeling

In addition to histone and genomic alterations, the impact of chromatin remodeling also plays a critical role in drug resistance. Remodeling entails the assembly of chromatin complexes into a high-order chromatin structure. These high-order chromatin structures can impact upon drug resistance, depending on whether the resulting accessibility permits transcription of drug-resistant signaling pathways. For instance, poly (ADP-ribose) polymerase 1 (PARP1), a critical enzyme involved in chromatin remodeling mechanisms, has emerged as an attractive target, and there are currently two FDA-approved PARP inhibitors, Oliparib and Veliparib [139].

Many studies have examined the role of chromatin remodeling on adaptive resistance evolution in GBM. In response to targeted kinase inhibitors, a small subset GBM stem cells will transition towards a survival adaptive state that is associated with the upregulation of histone demethylases KDM6A/B and widespread chromatin remodeling from disruptions of H3K27 trimethylated genomic regions [16]. Additional support for chromatin remodeling in GBM resistance to tyrosine-kinase pathway inhibitors has emerged from examining components of the tumor suppressor SWItch/Sucrose Non-Fermentable (SWI/SNF) complex which is critical for chromatin remodeling [140]. The Brahma-related gene 1 (BRG1) subunit of the SWI/SNF complex was implicated in GBM stemness and genetic/pharmacological of BRG1 reversed GBM stemness and enhanced sensitivity to alkylating chemotherapy [140]. Similar observations were noted by Hiramatsu and colleagues that SWI/SNF core complex plays essential roles in stemness maintenance in GSCs through association with a corepressor complex involving the d4-family proteins, DPF1/DPF3a [141].

Further evidence that chromatin remodeling is a critical mechanism of TMZ drug resistance in GBM has emerged from evaluating recurrent GBM tumors. Bruns et al. recently showed that chromatin remodeling influenced the transcription programs in paired recurrent versus newly diagnosed GBM patient tissues [142]. Specifically, treatment-induced alterations in GBM sensitivity are mediated through differential transcriptional hierarchies influenced by the chromatin remodeling machinery [142].

Chromatin remodeling is undoubtedly an integral part of therapeutic resistance evolution in GBM. Given the complexities of high-order chromatin structures, it is not surprising that targeting chromatin remodeling has been a challenge. Nevertheless, PARP inhibitors are currently in trial for GBM [70,71,72,73,143]. It is anticipated that anti-chromatin remodeling agents such as PARP inhibitors will overcome GBM resistance through preventing dynamic chromatin rearrangements.

### 3.4. Long Non-Coding RNAs

Long non-coding RNAs (lncRNAs) represent RNA molecules with a length of more than 200 nucleotides, which do not encode proteins [144]. These RNA molecules demonstrate widespread expression and impact on gene expression through interactions with the cellular epigenetic machinery. LncRNAs can therefore affect chromatin remodeling and other chromatin-associated functions. Structurally, lncRNAs retain 5-cap and alternative splicing features, but lack functional open reading frames (ORFs) required to encode proteins [145]. LncRNAs are dysregulated in multiple cancer types, including GBM [146,147]. By interacting with the epigenetic machinery in cancer cells, lncRNAs contribute to malignant neoplastic phenotypes such as metastasis, proliferation, and therapeutic resistance [148] (Figure 4).

The role of lncRNAs in GBM therapeutic resistance has recently emerged as an area of great interest and active investigation. Advances in bioinformatic approaches have permitted large scale analysis of differential expression of lncRNAs in treatment-naïve versus treatment resistant GBMs as well as comparisons between GBM and normal brain. In a recent study, approximately 300 lncRNAs were reported to be differentially expressed in TMZ resistant GBM patient tissues [149]. Furthermore, TMZ resistant GBMs had a propensity for dysregulation of lncRNAs, lending support for a potential role of lncRNAs in GBM drug resistance [149]. Besides their role in therapeutic resistance, lncRNAs have also been implicated in oncogenic signaling in GBM. Several studies have provided a nexus between oncogenic signaling and treatment resistance in GBM [150,151,152,153,154].

Metastasis-associated lung adenocarcinoma transcript 1 (MALAT1) is an oncogenic lncRNA that is highly expressed in GBM and that drives tumorigenesis and tumor propagation through regulations of miR-129, SOX2, and non-canonical Wnt signaling [152,154]. Furthermore, dysregulation of MALAT1 has been implicated as a contributor towards TMZ resistance of GBM in several studies [150,151,153,155,156]. Voce and colleagues examined how activation of MALAT1 during TMZ treatment of GBM cell lines contributed to TMZ resistance [153]. The investigators demonstrated that MALAT1 expression was induced through nuclear factor kappa-light-chain-enhancer of activated B cells (NF-κB) and p53 signaling [153]. They further showed that MALAT1 depletion restored TMZ sensitivity in patient-derived GBM cells both in vitro and in vivo [153]. In another study where MALAT1 was reported to be upregulated in GBM during TMZ resistance, depletion of MALAT1 restored TMZ sensitivity through mechanistic upregulation of miR-101 and downregulation of GSK-3β in resistant GBM cells [150]. Besides TMZ sensitivity, genetic silencing of MALAT1 significantly attenuated oncogenic phenotypes of cell growth, motility, and stemness in GBM [151,156]. Others have postulated that treatment-induced upregulation of *MALAT1* promotes GBM chemoresistance and oncogenic proliferation through suppression of miR-203 [155]. The above studies implicate MALAT1 as a facilitator of chemo-resistance in GBM and therefore a potential target for chemosensitization. These encouraging findings pave the way for future directions in developing novel strategies to overcome TMZ resistance in GBM.

H19 is another oncogenic lncRNA, whose upregulation was associated with GBM cell invasion, angiogenesis, and neurosphere formation [157]. Jiang and colleagues established the correlation between H19 expression and TMZ drug resistance in GBM, by demonstrating that H19 silencing decreased the IC-50 against TMZ and significantly increased apoptosis in GBM cells [158]. Duan and colleagues confirmed the above observations and demonstrated that H19 was upregulated through oxidative stress, and H19 expression contributed to TMZ resistance through NF-κB signaling activation [159]. Moreover, in addition of restoration of TMZ sensitivity, silencing of H19 leads to the reversal of epithelial–mesenchymal transition (EMT) in GBM cell lines through upregulation of E-cadherin and downregulation of Vimentin and zinc finger E-box binding homeobox 1 (*ZEB1*) expression [160]. These findings provide rationale for future investigations on targeting the oncogenic role of H19 in GBM as a novel avenue to overcome GBM drug resistance.

The Hox transcript antisense intergenic RNA (HOTAIR) is another lncRNA whose expression is significantly upregulated in GBM cells and plays a critical role in cell cycle progression through mechanistic binding to the 5′ domain of the polycomb repressive complex 2 (PRC2) [161]. In terms of TMZ resistance in GBM, it appears that HOTAIR modulates abnormal expression of Hexokinase 2 (HK2), a regulator of glycolysis and chemo-resistance. It was recently demonstrated that depletion of HOTAIR in GBM cells could suppress HK2 expression in protein and mRNA levels by targeting miR-125, therefore inhibiting cell proliferation and further enhancing the cytotoxicity of TMZ both in vitro and in vivo [162]. The above observations support a functional and targetable role for HOTAIR in GBM glycolytic metabolism and chemo-resistance reprogramming.

SOX2 Overlapping Transcript (SOX2OT) is an oncogenic lncRNA implicated in GBM cell proliferation and TMZ resistance [163]. SRY-Box Transcription Factor 2 (SOX2) is a transcription factor regulated by SOX2OT, and both SOX2 and SOX2OT are significantly upregulated in TMZ-resistant GBM cells [163]. Mechanistically, SOX2OT modulates TMZ sensitivity through demethylation of the SOX2 promoter and subsequent upregulation of SOX2 expression as well as activation of the Wnt5a/b-catenin pathway [163]. The downstream impact of SOX2OT activation cascade entails apoptosis inhibition, tumor proliferation, and TMZ resistance. Efforts to exploits vulnerabilities in SOX2OT/SOX2/Wnt5a/b-catenin signaling cascade provides a novel avenue for targeting TMZ resistance in GBM.

FOXD2 Adjacent Opposite Strand RNA 1 (FoxD2-AS1) is an oncogenic lncRNA that is particularly interesting given its potential impact on *MGMT* methylation in GBM. As previously alluded, *MGMT* methylation status is a strong predictor of TMZ sensitivity in GBM whereby hypomethylation promotes TMZ resistance and hypermethylation enhances TMZ sensitivity. Recently, it was reported that FoxD2-AS1 expression was correlated with worse outcomes for GBM patients, and FoxD2-AS1 was responsible for GBM malignant transformation and TMZ resistance [164]. Furthermore, high FoxD2-AS1 expressions correlated with hypomethylation of the *MGMT* promoter in GBM patients [164]. Interestingly, genetic silencing of FoxD2-AS1 rendered the *MGMT* promoter hypermethylated and restored TMZ sensitivity in GBM [164]. These observations provide valuable insights on restoring TMZ sensitivity through regulation of *MGMT* promoter.

There is clearly mounting evidence from multiple studies that lncRNAs play an important role in GBM drug resistance through a variety of mechanisms (Figure 4). It is also apparent that lncRNAs activity can be modulated to successfully reverse GBM malignant phenotypes and restore sensitivity to TMZ. Further research is necessary for optimal and selective targeting of lncRNAs to overcome GBM resistance while simultaneously minimizing unwanted global effects.

## 4. Conclusions

GBM remains one of the most lethal cancers despite a significant scientific advance in our understanding of the molecular underpinnings of GBM tumorigenesis. There are significant challenges in developing impactful therapies to improve the outcomes of GBM patients. So far, TMZ plus RT remains the most effective therapy, yet the durability of response is often short-lived secondary to resistance evolution largely driven through epigenetic mechanisms. To generate impactful therapies that could improve outcomes for GBM patients, contextual understanding and targeting of treatment-induced epigenetic modifications is paramount. It will therefore be important that future studies focus on elucidating epigenetic targets of divergent evolution in GBM stem cells using appropriate models.

To date, epigenetic drugs have not significantly improved survival in GBM. Furthermore, the global effects of these drugs on epigenetic modifications have resulted in significant toxicities, thereby limiting tolerance to treatment. Therefore, future efforts should be directed towards rational epigenetic targeting to improve TMZ response while simultaneously minimizing systemic toxicity.

## Figures and Tables

**Figure 1 ijms-22-08324-f001:**
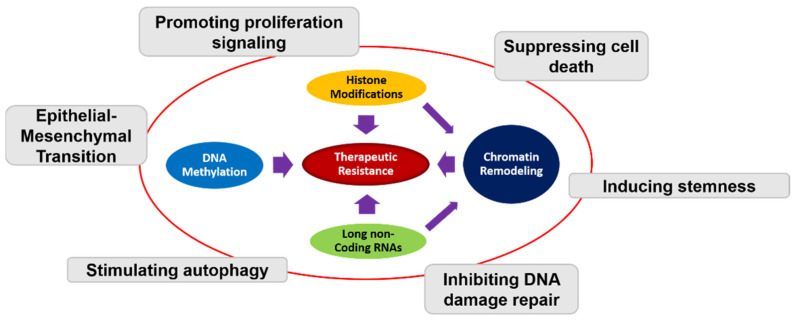
Schematic representation of the interplay between different epigenetic mechanisms and therapeutic resistance in GBM. DNA methylation, histone modifications, chromatin remodeling, and long non-coding RNAs all contribute to therapeutic resistance through different mechanism including promoting proliferation, suppressing cell death, inducing stemness, inhibiting DNA damage repair, stimulating autophagy, and epithelial–mesenchymal transition (EMT).

**Figure 2 ijms-22-08324-f002:**
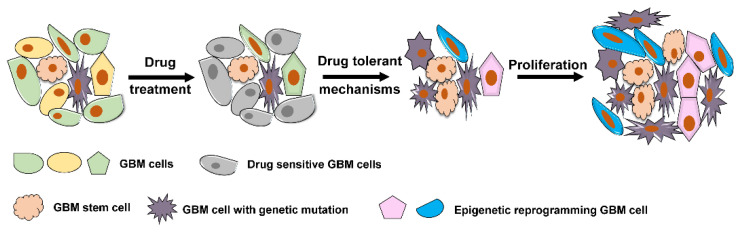
Representative mechanism underlying drug resistance in GBM. GBM cells that carry resistance-conferring mutation(s) in a heterogeneous cancer population and GBM stem cells that are intrinsically resistant to drug treatment survive drug exposure and outgrow to determine the further cancer population. A group of GBM cells evolved through epigenetic reprogramming become drug-tolerant cells and further confer resistance to drug therapy.

**Figure 3 ijms-22-08324-f003:**
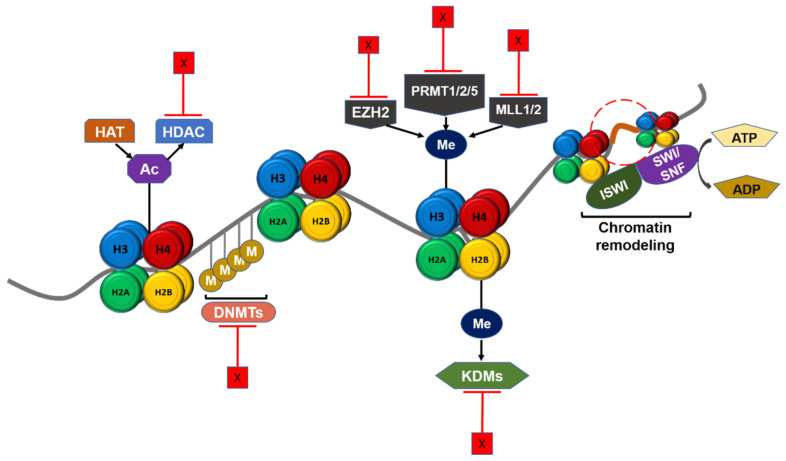
Schematic summary of epigenetic regulation in GBM and potential therapeutic targets. Chromatin structure and gene transcription are regulated by epigenetic mechanisms. DNA methyltransferase (DNMT), histone lysine acetyltransferase (HAT), and histone lysine methyltransferase (KMT) such as EZH2, PRMTs, and MLL1/2 catalyze the addition of epigenetic groups to either DNA or histone tails. Histone deacetylase (HDAC) and histone demethylase (KDM) remove these epigenetic groups. ISWI and SWI/SNF complex could remodel chromatin structure according to histone modifications. These regulators can be targeted in therapeutic treatment against GBM, and furthermore the targets shown here are either under pre-clinical studies or approved and under clinical trials. Ac: Acetyl group. M/Me: Methyl group. X: Inhibition.

**Figure 4 ijms-22-08324-f004:**
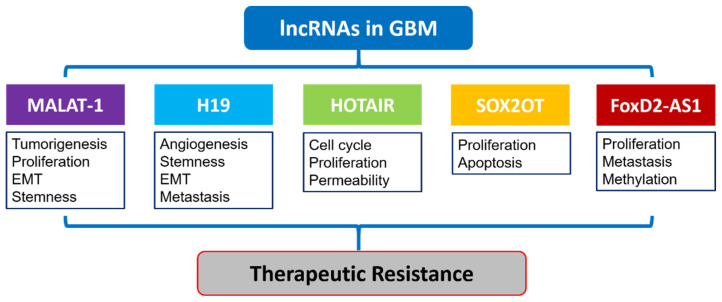
Schematic summary of several important lncRNAs in therapeutic resistance of GBM. The lncRNAs MALAT-1. H19, HOTAIR, SOXOT, and FoxD2-AS1 are involved in the therapeutic resistance of GBM through different mechanisms including tumorigenesis, proliferation, EMT, angiogenesis, stemness, metastasis, cell cycle, and apoptosis.

**Table 1 ijms-22-08324-t001:** Development of combination therapy using epigenetic drugs for the treatment of GBM.

Clinical Trials Identifier	Clinical Trial Phase	Intervention/Treatment	Condition	Status	Primary Endpoint	Secondary Endpoint	References
NCT00268385	Phase I	Vorinostat + TMZ	GBM	Active	MTD (maximum tolerated dose) of Vorinostat with TMZ	1. Efficacy in terms of anti-tumor activity based on clinical, radiographic, and biologic assessments 2. Plasma pharmacokinetic parameters of Vorinostat	[51]
NCT00731731	Phase I/II	Vorinostat + TMZ + RT	Newly diagnosed GBM	Active	1. MTD of Vorinostat (Phase I) 2. Overall Survival (OS) (Phase II)	1. Time to tumor progression (Phase II) 2. Incidence of adverse events (Phase II)	[52]
NCT00555399	Phase I/II	Vorinostat + TMZ + isotretinoin	Recurrent GBM	Completed	MTD		[53]
NCT01738646	Phase II	Vorinostat + Bevacizumab	Recurrent GBM	Completed	Six-month progression-free survival (PFS)	1. Radiographic response 2. Median PFS 3. Median OS	[54]
NCT01266031	Phase I/II	Vorinostat + Bevacizumab	Recurrent GBM	Completed	1. PFS at 6 Months 2. MTD of Oral Vorinostat used with Bevacizumab	1. Time to Tumor Progression 2. OS 3. Effects of Bevacizumab with and without Vorinostat upon biomarkers 4. Mean symptom interference at the time of clinical evaluation 5. Radiological response	[55]
NCT00762255	Phase I	Vorinostat + Bevacizumab + irinotecan	Recurrent GBM	Completed	MTD	1. PFS at 6 months 2. Number of participants with adverse events	[56]
NCT00939991	Phase I/II	Vorinostat + TMZ + Bevacizumab	Recurrent GBM	Completed	1. Determination of MTD (Phase I) 2. 6-month PFS (Phase II)	1. Radiographic response (Phase II) 2. PFS (Phase II) 3. OS (Phase II) 4. Number of patients with Grade 2 or greater, treatment-related toxicities (Phase II)	[57]
NCT01236560	Phase II/III	Vorinostat + TMZ + Bevacizumab	Young newly diagnosed GBM	Active	1. MTD of Vorinostat 2. Event-free survival	1. OS 2. Cumulative incidence of disease progression in each treatment arm	[58]
NCT00641706	Phase II	Vorinostat + Bortezomib	Recurrent GBM	Completed	PFS at 6 Months	1. OS 2. Time to tumor progression 3. Proportion of confirmed tumor response	[59]
NCT01110876	Phase I/II	Vorinostat + TMZ + Enlotinib	Recurrent GBM	Terminated	MTD of Vorinostat in combination with escalating doses of erlotinib and TMZ	PFS	[60]
NCT03426891	Phase I	Vorinostat + Pembrolizumab + TMZ + RT	GBM	Active	MTD	OS	[61]
NCT02137759	Phase II	Belinostat + TMZ + RT	Newly diagnosed GBM	Active	1. PFS 2. MTD	1. PFS 2. OS 3. IDS-SR score change	[62]
NCT00302159	Phase II	Valproic acid + TMZ + RT	GBM	Completed	1. Median PFS 2. Percentage of participants with PFS at 6, 12, and 24 months 3. Number of participants with best response 4. Median OS 5. Percentage of participants with OS at 6, 12, and 24 months	Number of participants with adverse events	[63]
NCT00879437	Phase II	Valproic acid + Bevacizumab + RT	GBM	Terminated	One-year event-free survival	1. Median EFS 2. Median OS 3. Partial response in diffuse intrinsic pontine Glioma 4. Partial response in high-grade Gliomas 5. Complete response in high-grade Gliomas	[64]
NCT02648633	Phase I	Valproic acid + Nivolumab + RT	GBM	Completed	1. Feasibility based on number of subjects who complete 4 doses of nivolumab 2. Incidence of adverse events	1. Clinical response rate 2. Incidence of pseudoprogressions	[65]
NCT01817751	Phase II	Valproic acid + sorafenib tosylate + sildenafil citrate	GBM	Active	PFS	1. Overall best response rate 2. OS 3. Incidence of adverse events	[66]
NCT03243461	Phase III	Valproic acid + TMZ	GBM	Active	Comparison of effects of Valproine acid with respect to historical control group		[67]
NCT00859222	Phase I/II	Bevacizumab + Panobinostat	Recurrent GBM	Completed	1. LBH589 MTD (Phase I) 2. Dose limiting toxicity (Phase I) 3. PFS at 6 months (Phase II)	1. Best radiographic response 2. PFS (Phase II) 3. OS (Phase II)	[68]
NCT03684811	Phase I/II	Azacitidine + FT-2102	GBM	Active	1. Number of participants with dose limiting toxicity (Phase 1) 2. Doses recommended for future studies (Phase 1) 3. Objective response rate of FT-2102 single agent or in combination with Azacitidine (Phase 2)	Phase 1 and 2: 1. OS 2. Time to response (TTR) 3. Time to tumor progression 4. Duration of response (DOR) 5. PFS 6. Drug level within CSF	[69]
NCT04614909	Early Phase I	Olaparib + TMZ + RT	Newly diagnosed GBM	Active	Systemic plasma PK profile parameters	1. PFS participants with demonstrated PK effects 2. OS 3. Drug-related toxicity 4. Adverse events 5. Treatment-emergent adverse events 6. Deaths	[70]
NCT02152982	Phase II/III	Veliparib + TMZ	Newly diagnosed GBM	Active	OS	1. Interaction with Optune device 2. PFS 3. Objective tumor response 4. Overall adverse event rates for grade 3 or higher adverse events 5. Change in quality of life (QOL)	[71]
NCT01026493	Phase I/II	Veliparib + TMZ	Recurrent GBM	Completed	1. MTD (Phase I) 2. 6-month PFS rate for patients with measurable disease after surgery (Phase II)	Phase II: 1. Objective response rate for patients with measurable disease after surgery 2. OS	[72]
NCT03581292	Phase II	Veliparib + TMZ + RT	GBM	Active	Event-free Survival	1. Objective response 2. OS	[73]

## Data Availability

Not applicable.

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
