# Peer review of "The Impact of Epigenetic Modifications on Adaptive Resistance Evolution in Glioblastoma"

_ijms, 2021, doi:10.3390/ijms22158324_

Round 1
Reviewer 1 Report
In this paper, the authors summarize the latest updates on the impact of epigenetic modifications on adaptive resistance evolution in GBM to therapy.
GBM is one of the most lethal cancers although significant scientific advances.
There are major challenges in developing impactful therapies to improve the outcomes of GBM patients.
Therefore, the topic handled in this work is important. Also, the paper has been well-organized and written carefully.
Contextual understanding and targeting of treatment-induced epigenetic modifications are important to generate impactful therapies and to improve outcomes for GBM patients.
Therefore, this is still an active research area and this survey can be helpful for many researchers interested in this area.
Reviewer 2 Report
- The manuscript is scientifically sound and very well written
- The abstract is well written
- I recommend avoid using abbreviations in Figure 1 (such as EMT).
- The manuscript could benefit from very minor editing for grammar, missing words, and subject-verb agreement, etc. It is recommended that authors delete irrelevant "general" phrases and sentences, repeated and unneeded words. They should use short sentences. For example, the subheading “Epigenetic alteration involVed in drug resistance of GBM” should be corrected “Epigenetic alteration involved in drug resistance of GBM”.
- Table 1: I suggest adding the primary and secondary end points of each clinical trial.
- All abbreviations should be revised and defined at their first use.
